# The Significance of Prostate Specific Antigen Persistence in Prostate Cancer Risk Groups on Long-Term Oncological Outcomes

**DOI:** 10.3390/cancers13102453

**Published:** 2021-05-18

**Authors:** Daimantas Milonas, Zilvinas Venclovas, Gustas Sasnauskas, Tomas Ruzgas

**Affiliations:** 1Medical Academy, Department of Urology, Lithuanian University of Health Sciences, 44307 Kaunas, Lithuania; zilvinas.venclovas@stud.lsmu.lt (Z.V.); gustas.sasnauskas@stud.lsmu.lt (G.S.); 2Department of Applied Mathematics, Kaunas University of Technology, 44249 Kaunas, Lithuania; tomas.ruzgas@ktu.lt

**Keywords:** prostate cancer, PSA persistence, risk groups, radical prostatectomy, outcomes

## Abstract

**Simple Summary:**

The current prostate cancer guidelines recommend performing the first prostate-specific antigen measurement at three months after radical prostatectomy. However, at an earlier measurement, persistence (≥0.1 ng/mL) of this biomarker could be found in up to 30% of cases, depending on the prostate cancer risk factors. Recent reports have demonstrated an increasing interest in prostate-specific antigen persistence as a possible additional predictor of disease progression and cancer-specific survival. However, the data remain scant, with weak evidence. We assessed the relationship between prostate-specific antigen persistence and long-term oncological outcomes within prostate cancer risk groups. We found that persistence of this biomarker could be used as an independent predictor of worse long-term outcomes in high-risk prostate cancer patients, while in intermediate-risk patients, this parameter significantly predicts only biochemical recurrence and has no impact on the outcomes in low-risk patients.

**Abstract:**

Objective: To assess the significance of prostate-specific antigen (PSA) persistence at the first measurement after radical prostatectomy (RP) on long-term outcomes in different prostate cancer risk groups. Methods: Persistent PSA was defined as ≥0.1 ng/mL at 4–8 weeks after RP. Patients were stratified into low-, intermediate- and high-risk groups, according to the preoperative PSA, pathological stage, grade group and lymph nodes status. The ten-year cumulative incidence of biochemical recurrence (BCR), metastases, cancer-specific mortality (CSM) and overall mortality (OM) were calculated in patients with undetectable and persistent PSA in different PCa-risk groups. Multivariate regression analyses depicted the significance of PSA persistence on each study endpoint. Results: Of all 1225 men, in 246 (20.1%), PSA persistence was detected. These men had an increased risk of BCR (hazard ratio (HR) 4.2, *p* < 0.0001), metastases (HR: 2.7, *p* = 0.002), CRM (HR: 5.5, *p* = 0.002) and OM (HR: 1.8, *p* = 0.01) compared to the men with undetectable PSA. The same significance of PSA persistence on each study endpoint was found in the high-risk group (HR: 2.5 to 6.2, *p* = 0.02 to *p* < 0.0001). In the intermediate-risk group, PSA persistence was found as a predictor of BCR (HR: 3.9, *p* < 0.0001), while, in the low-risk group, PSA persistence was not detected as a significant predictor of outcomes after RP. Conclusions: Persistent PSA could be used as an independent predictor of worse long-term outcomes in high-risk PCa patients, while, in intermediate-risk patients, this parameter significantly predicts only biochemical recurrence and has no impact on the outcomes in low-risk PCa patients.

## 1. Introduction

Radical prostatectomy (RP) with or without lymph node dissection is an accepted treatment modality in patients with localized and locally advanced prostate cancer (PCa) [1,2,3]. However, a non-negligible number of patients may experience disease recurrence following RP. The European Association of Urology (EAU) prostate cancer (PCa) guidelines recommend performing the first prostate-specific antigen (PSA) measurement at three months after RP [4]. Despite that, at early PSA measurements within 4–8 weeks, PSA persistence (≥0.1 ng/mL) could be found in 8–26% of men, depending on the PCa risk factors [5,6]. Recent reports have demonstrated an increasing interest in PSA persistence as a possible additional predictor of disease progression and cancer-specific survival [5,6,7,8,9,10,11,12,13]. A very recent systemic review and a meta-analysis included studies assessing the importance of PSA persistence on oncological outcomes after RP and the management of persistently elevated PSA [14,15]. The authors demonstrated the association between PSA persistence and biochemical recurrence (BCR), disease progression and cancer-specific mortality (CSM) and suggested a benefit from immediate radiotherapy [15]. The main limitation of this analysis is the low number of studies analyzing the endpoints of different outcomes. Therefore, to date, there is a lack of evidence on the prognostic significance of PSA persistence. Despite presenting, in several reports, a strong association between PSA persistence and unfavorable cancer characteristics [5,7,11], there are no analyses on the importance of PSA persistence in different PCa-risk groups.

The aim of our study was to assess the relationship between persistent PSA at the first measurement between 4 and 8 weeks after RP and the long-term oncological outcomes within the PCa-risk groups. We hypothesized that the significance of PSA persistence on the oncological outcome differed in the PCa-risk groups. Retrospective analyses were performed assessing the significance of PSA persistence on the 10-year cumulative incidence of BCR, metastases (MTS), CSM and overall mortality (OM) in patients after RP.

## 2. Material and Methods

### 2.1. Patient Population

Between 2001 and 2019, 2611 men were treated by RP for clinically localized and locally advanced PCa at the Department of Urology of the Lithuanian University of Health Sciences. Patient data were registered in a PCa database. During the study period, RP was performed by 9 senior urologists. The exclusion criteria were as follows: no follow-up data, a patient did not have PSA measurement within the first 2 months after RP, neo- and adjuvant treatments, detected metastases before RP and incomplete clinical or pathological data. Patients were stratified according to undetectable PSA (PSA < 0.1 ng/mL) vs. persistent PSA (PSA ≥ 0.1 ng/mL) at the first measurement 4–8 weeks after RP. Furthermore, patients were divided according to the pathological PCa features into low-risk (pT2, GG1, PSA <10 ng/mL and N_x_, pN0); intermediate-risk (pT3a, GG2–3, PSA 10–20 ng/mL and N_x_, pN0) and high-risk (pT3b, GG4–5, PSA > 20 ng/mL and pN1) groups. Our study flowchart is presented in Figure 1. Tumor grading was classified using the revised 2005 International Society of Urologic Pathology (ISUP) Gleason score grading system and the suggested the new GG model after 2014 [16,17]. The university’s ethical committee approved the collection and analysis of the data.

### 2.2. Outcomes

The study endpoints were the 10-year cumulative incidences of BCR, MTS, CSM and OM. BCR was defined as PSA > 0.2 ng/mL at two consecutive measurements. MTS were defined as skeletal or visceral lesions confirmed by a bone scan, computed tomography (CT) or magnetic resonance imaging (MRI) using RECIST criteria. Local and loco-regional recurrence was histopathologically confirmed by surgery or biopsy or by MRI. MTS-free survival was defined as the time from surgery until the detection of MTS. Death without MTS was considered a competing event. Alive patients without MTS were censored at the last follow-up. OM was defined as the time from surgery until death for any cause. Alive patients were censored. CSM was defined as the time from surgery until cancer-related death. Death for other causes was considered a competing event. Alive patients were censored as well. Data about patient death were taken from the national healthcare database. All cases of death were rechecked with follow-up data available in the center database.

### 2.3. Statistical Analysis

Descriptive statistics included frequencies and proportions for categorical variables, medians with interquartile range and means with 95% confidence intervals (CI) for continuous variables. The chi-square and Mann–Whitney *U* tests were used to assess the differences between groups. The cumulative incidence function was used to estimate the 10-year BCR, MTS, CSM and OM in men with persistent vs. undetectable PSA in different PCa-risk groups. The Kaplan–Meier method was used to estimate survival in the study groups. A multivariate logistic regression model was used to test the relationship between the clinicopathological covariates and study endpoints. The results were presented as hazard ratios (HR) with 95% CI. Analyses were performed using SAS software (version 9.4 of the SAS System for Windows, by SAS Institute Inc., Cary, NC, USA) with the two-sided significance level set at *p* < 0.05.

## 3. Results

The patient characteristics are presented in Table 1. The median follow-up (interquartile range) was 103 (IQR 50–157) months. Of all 1225 studied men, 246 (20.1%) had PSA persistence at the first measurement within 4–8 weeks after RP. PSA persistence was detected in 23 of 261 (8.8%), 94 of 705 (13.3%) and 129 of 259 (49.8%) men in the low-, intermediate- and high-risk groups, respectively. Overall, the 10-year cumulative incidence of BCR, MTS, CSM and OM were 39.61% (95% CI: 35.95–43.64), 9.70% (95% CI: 7.67–12.27), 4.81% (95% CI: 3.49–6.61) and 18.15% (95% CI: 16.04–20.53), respectively, with a significantly higher incidence in men with PSA persistence (all *p* < 0.0001, Figure 2). In the multivariate regression analysis, PSA persistence was detected as an independent predictor for each study endpoint (HR: 1.8 to 5.5, *p* < 0.01 to *p* < 0.0001, see Table 2).

### 3.1. Outcomes in the Low-Risk Group

The mean persistent PSA value at the first measurement was 0.31 ng/mL (95% CI: 0.16–0.47), and the range was 0.1–1.4 ng/mL. At the 10-year cumulative incidence of BCR, MTS, CSM and OM, the rates in men with undetectable vs. persistent PSA were 11.3% vs. 32.9% (*p* = 0.03), 0.6% vs. 0.8% (*p* = 0.4), 1.0% vs. 4.8% (*p* = 0.2) and 14.1% vs. 26.6% (*p* = 0.07), respectively (Figure 3). In the multivariate regression analysis, PSA persistence was not detected as a significant predictor of BCR, metastases, CSM or OM (Table 3).

### 3.2. Outcomes in the Intermediate-Risk Group

The mean persistent PSA value in this group was 0.32 ng/mL (95% CI: 0.23–0.41), and the range was 0.1–3.0 ng/mL. Comparing men with undetectable vs. persistent PSA, the 10-year cumulative incidence of BCR was 31.3% vs. 80.8% (*p* < 0.0001); the incidence of MTS was 4.7% vs. 13.7% (*p* = 0.03), CSM was 1.35% vs. 4.8% (*p* = 0.09) and OM was 14.8% vs. 18.7 % (*p* = 0.2) (Figure 4). The multivariate regression analysis revealed that PSA persistence was a significant predictor of BCR (HR: 3.8, *p* < 0.0001), while significance was not detected in the analyses of MTS (HR: 2.6, *p* = 0.1), CSM (HR: 4.2, *p* = 0.2) and OM (HR: 1.03, *p* = 0.9) (Table 3).

### 3.3. Outcomes in the High-Risk Group

The mean persistent PSA in this group was 1.9 ng/mL (95% CI: 1.29–2.50), and the range was 0.1–24.6 ng/mL. The cumulative 10-year incidence of BCR, MTS, CSM and OM was significantly lower in men with undetectable PSA compared to men with PSA persistence: 50.3% vs. 98% (*p* < 0.0001), 12.8% vs. 49% (*p* < 0.0001), 7% vs. 31.9% (*p* < 0.0001) and 21.2% vs. 43.6% (*p* = 0.0001), respectively (Figure 5). Differently to other study groups, men with detected PSA persistence had a significantly increased risk of BCR (HR: 5.1, *p* < 0.0001), risk of MTS (HR: 2.6, *p* = 0.015), CSM (HR: 6.2, *p* = 0.007) and OM (HR: 2.7, *p* = 0.005) in the multivariate analysis (Table 3).

## 4. Discussion

PSA remains the most important surrogate biomarker in the follow-up after radical PCa treatment. Recent studies have demonstrated the importance of PSA persistence in disease progression and CSM after RP [14,15]. However, published data are scant, and there are no reports analyzing the importance of PSA persistence according to PCa risk groups. We investigated the relationship between PSA persistence and long-term oncological outcomes in patients with different risk groups.

Among all >1200 patients with available first PSA measurements within 4–8 weeks after RP, we found that PSA persistence (>0.1 ng/mL) was a strong predictor of BCR, MTS, CSM and OM. When patients with low-, intermediate- and high-risk PCa features were analyzed separately, we detected different predictive probabilities of PSA persistence for the study endpoints. The results presented herein could provide important prognostic information for clinicians and patients very shortly after the initial surgical treatment. A major implication of our study is that the early detection of PSA failure after the initial surgical treatment may assist in more personalized follow-up and additional treatment decision-making.

In the low-risk patient cohort, PSA persistence was associated with a worse 10-year cumulative incidence of BCR (33% vs. 11%, *p* = 0.03) compared to men with undetectable PSA. However, in the multivariate regression analysis, PSA persistence was not found as a significant predictor of BCR. Moreover, in low-risk patients, PSA persistence was not associated with a more rapid disease progression and mortality. Our results demonstrate that men with favorable cancer features and PSA persistence will very likely have the same low risk of developing MTS and cancer-related death as men with undetectable PSA. Despite the association with an increased incidence of BCR, PSA persistence in low-risk PCa should not be used as an indicator for early salvage treatment.

In the intermediate-risk group, men with PSA persistence demonstrated an increased 10-year cumulative incidence of BCR and MTS, with absolute 51% and 8% differences compared to men with undetectable PSA (*p* < 0.0001 and *p* < 0.02, respectively), while the regression analysis revealed that PSA persistence could predict only BCR (HR: 3.9, *p* < 0.0001). In intermediate-risk patients, PSA persistence demonstrated a significant association with BCR. However, this association did not translate to a significantly higher risk of MTS or CSM. Probably, these men are good candidates for an intensified PSA follow-up and the initiation of additional delayed treatment, according to the PSA dynamics [18], rather than early salvage treatment [19].

Differently to the other groups, in the patients with high-risk PCa features, PSA persistence was associated with a significantly higher 10-year cumulative incidence of BCR, MTS, CSM and OM (absolute differences 48%, 39%, 26% and 24%, respectively). Moreover, PSA persistence was found as a significant independent predictor at each study endpoint (HR: 2.5–6.2, *p* < 0.02 to <0.0001). The aggressive nature of high-risk PCa is well-known from previous studies [1]. However, this specific PCa subset is heterogeneous with different risks of progression and responses to salvage treatments [20,21]. An additional validated biomarker would be very helpful for the recognition of the most aggressive high-risk prostate cancer. The results of the current study show that men with high-risk PCa features and PSA persistence are at a significantly higher risk of worse outcomes compared to men with high-risk cancer and undetectable PSA. Moreover, the regression analysis demonstrated that PSA persistence is the most important predictor of CSM. Therefore, PSA persistence could be useful in clinical practice for the early identification of patients with potentially rapid disease progression after RP. Similarly, a higher PSA level at 3 months after radiation therapy was found as a significant predictor of worse outcomes, mainly in high-risk patients [22].

Taken together, the presented study results suggest that the importance of PSA persistence reported in previous studies [5,6,7,8,9,10,11,12,13,14] may not be directly translated to each patient: the significance of PSA persistence is marginal in low-risk patients, and PSA persistence has the biggest impact on the outcomes in high-risk PCa patients. Patient stratification according to unfavorable cancer features is essential for the correct interpretation of PSA persistence.

The current study is not devoid of limitations. The retrospective study design and unavailable PSA data within 4–8 weeks after RP may have created a selection bias. The single-center database and relatively low number of final events, especially in the low- and intermediate-risk groups, could influence the significance of the presented results. Only men after RP were included into the analysis, while a direct comparison of our findings with early PSA detection after radiation therapy could confirm the importance of the presented results.

The results of the current study demonstrate the incidence, as well as value, of a persistent PSA increase with a higher PCa-risk group. PSA persistence was associated with a worse 10-year cumulative incidence of BCR, MTS, CSM and OM compared to men with undetectable PSA when analyzing the whole cohort. However, in low-risk patients, a significant survival difference was found for BCR; in the intermediate-risk, for BCR and MTS and only in high-risk patients for each outcome. We noted that, in the regression analyses, the predictive probability of PSA persistence was very similar in the high-risk group and in all the study population. Such findings suggest that the importance of PSA persistence should be interpreted differently in PCa-risk groups, with the highest value of significance in cancer with unfavorable features.

## 5. Conclusions

Persistent PSA could be used as an independent predictor of worse long-term outcomes in the high-risk PCa patients, while in the intermediate-risk patients this parameter significantly predicts only biochemical recurrence and has no impact on outcomes in the low-risk PCa patients.

## Figures and Tables

**Figure 1 cancers-13-02453-f001:**
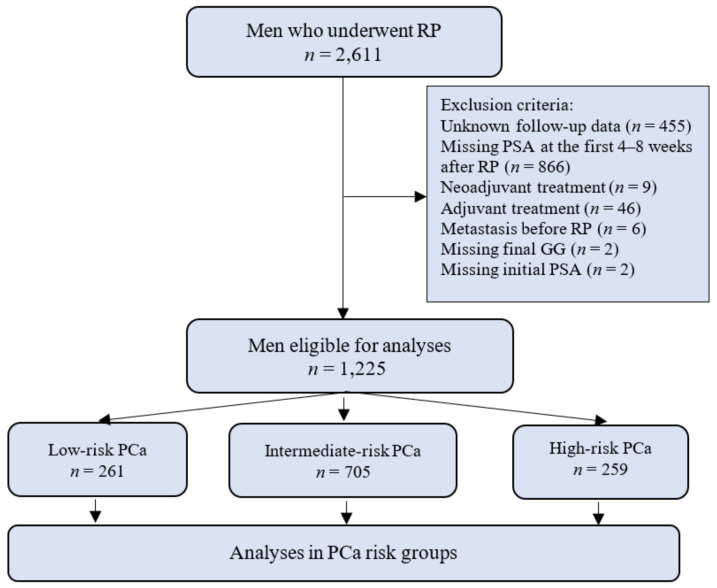
Study flowchart. RP—radical prostatectomy, PCa—prostate cancer, GG—grade groups and PSA—prostate-specific antigen.

**Figure 2 cancers-13-02453-f002:**
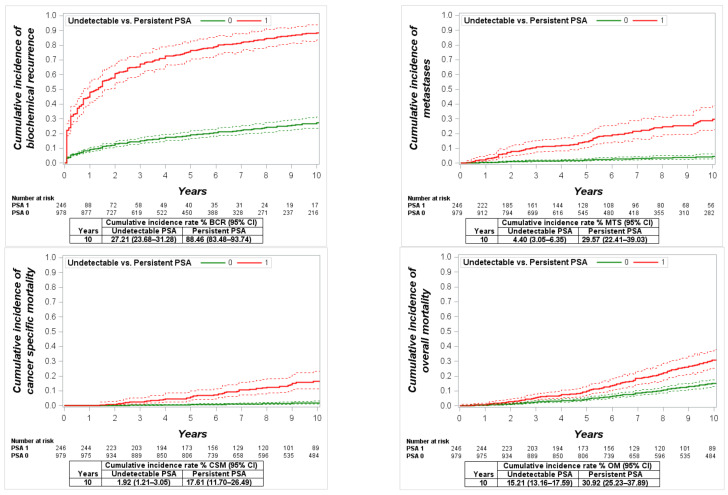
Cumulative incidence function for biochemical recurrence (BCR), metastases (MTS), cancer-specific mortality (CSM) and overall mortality (OM) in all study cohort patients with undetectable vs. persistent prostate-specific antigen (PSA).

**Figure 3 cancers-13-02453-f003:**
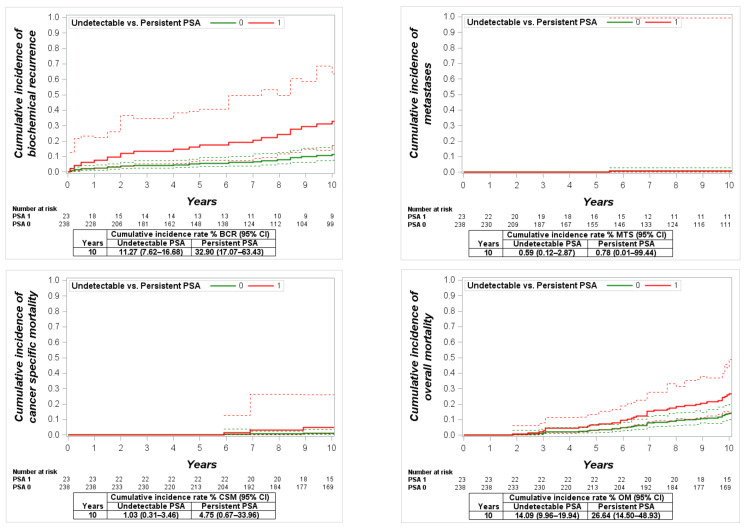
Cumulative incidence function for biochemical recurrence (BCR), metastases (MTS), cancer-specific mortality(CSM) and overall mortality (OM) in low-risk patients with undetectable vs. persistent prostate-specific antigen (PSA).

**Figure 4 cancers-13-02453-f004:**
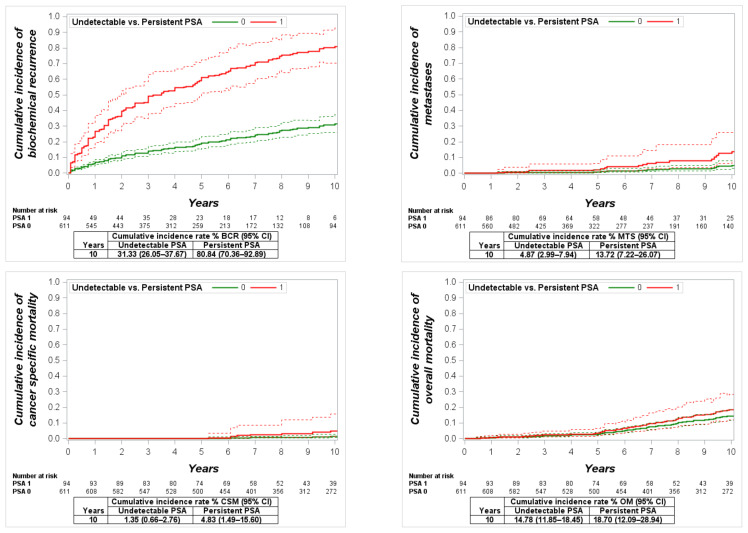
Cumulative incidence function for biochemical recurrence (BCR), metastases (MTS), cancer-specific mortality (CSM) and overall mortality (OM) in intermediate-risk patients with undetectable vs. persistent prostate-specific antigen (PSA).

**Figure 5 cancers-13-02453-f005:**
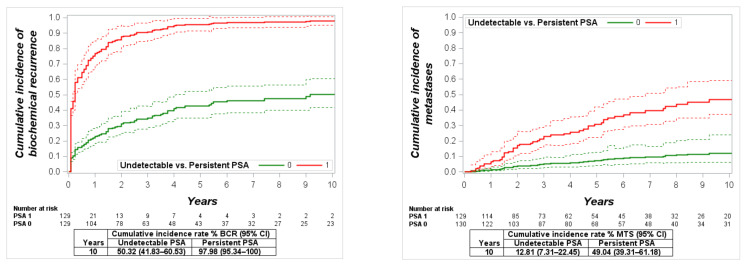
Cumulative incidence function for biochemical recurrence (BCR), metastases (MTS), cancer-specific mortality (CSM) and overall mortality (OM) in high-risk patients with undetectable vs. persistent prostate-specific antigen (PSA).

**Table 1 cancers-13-02453-t001:** Patient characteristics.

Parameter	All *n* = 1225	Low-Risk *n* = 261	Intermediate-Risk *n* = 705	High-Risk *n* = 259	*p*-Value
Age (years): median (IQR)	64 (59–68)	63.5 (59–68)	63 (58–68)	65 (60–69)	0.01
PSA (ng/mL): median (IQR)	6.5 (4.8–9.95)	55 (4.3–6.9)	6.3 (4.8–9.6)	10.8 (6.2–21.1)	<0.0001
Clinical stage: *n*, (%)					<0.0001
cT1	337 (27.5)	106 (40.6)	197 (27.9)	34 (13.1)	
cT2	706 (57.6)	144 (55.2)	428 (60.7)	134 (51.7)	
cT3	178 (14.5)	11 (4.2)	76 (10.8)	91 (35.1)	
Unknown	4 (0.3)	-	4 (0.6)	-	
Biopsy Gleason score: *n*, (%)					<0.0001
6	710 (58)	249 (95.4)	394 (55.9)	67 (25.9)	
3 + 4	363 (29.6)	11 (4.2)	270 (38.3)	82 (31.7)	
4 + 3	53 (4.3)	1 (0.4)	22 (3.2)	30 (11.6)	
8	64 (5.2)	-	16 (2.3)	48 (18.5)	
9–10	33 (2.7)	-	1 (0.1)	32 (12.4)	
Unknown	2 (0.2)	-	2 (0.2)	-	
Pathological Gleason Score: *n*, (%)					<0.0001
6	329 (26.9)	261 (100)	58 (8.2)	10 (3.9)	
3 + 4	626 (51.1)	-	569 (80.7)	57 (22)	
4 + 3	113 (9.2)	-	78 (11.1)	35 (13.5)	
8	70 (5.7)	-	-	70 (27.0)	
9–10	87 (7.1)	-	-	87 (33.6)	
Pathologic stage: *n*, (%)					<0.0001
pT2	776 (63.3)	261 (100)	459 (65.1)	56 (21.6)	
pT3a	326 (26.6)	-	246 (34.9)	80 (30.9)	
pT3b-pT4	123 (10)	-	-	123 (47.5)	
Positive surgical margin: *n*, (%)	399 (33.7)	41 (16.1)	221 (32.2)	137 (56.1)	<0.0001
PLND: *n*, (%)	489 (39.9)	61 (23.4)	228 (32.3)	200 (77.2)	<0.0001
LNI: *n*, (%)	65 (13.3)	-	-	65 (32.5)	<0.0001
Persistent PSA	246 (20.1)	23 (8.8)	94 (13.3)	129 (49.8)	<0.0001
BCR: *n*, (%)	383 (31.3)	27 (10.3)	179 (25.4)	177 (68.3)	<0.0001
MTS: *n*, (%)	87 (7.1)	4 (1.5)	24 (3.4)	59 (22.8)	<0.0001
Death: *n*, (%)	226 (18.4)	46 (17.6)	113 (16)	67 (25.9)	<0.0001
Cancer related death: *n*, (%)	45 (3.8)	3 (1.1)	11 (1.6)	33 (12.7)	<0.0001

IQR—interquartile range, PSA—prostate-specific antigen, LN—lymph nodes, LNI—lymph node invasion, BCR—biochemical recurrence and MTS—metastasis.

**Table 2 cancers-13-02453-t002:** Multivariable competing risk analysis of the biochemical recurrence (BCR), developing metastases (MTS), overall mortality (OM) and cancer-specific mortality (CSM) in the whole cohort.

Parameter	BCR			MTS			OM			CSM		
	HR	95% CI	*p*	HR	95% CI	*p*	HR	95% CI	*p*	HR	95% CI	*p*
PSA (ng/mL)	1.1	1–1.02	0.05	0.9	0.97–1.01	0.3	0.9	0.97–1.02	0.6	0.9	0.92–1.01	0.08
Age (years)	1.1	0.99–1.04	0.2	1.1	0.97–1.06	0.6	1.1	1.04–1.12	<0.001	1.1	1.0–1.14	0.04
PSA persistence	4.2	3.06–5.76	<0.001	2.7	1.44–5.09	0.002	1.8	1.13–2.76	0.01	5.5	2.08–14.49	0.006
Stage:												
pT2		Ref.			Ref.			Ref.			Ref.	
pT3a	1.7	1.19–2.55	0.004	0.7	0.33–1.54	0.4	0.8	0.48–1.44	0.5	0.7	0.21–2.38	0.6
pT3b-pT4	2.3	1.48–3.59	0.0002	1.4	0.62–2.97	0.5	1.7	0.89–3.07	0.1	2.5	0.84–7.4	0.09
Grade Group:												
GG 1		Ref.			Ref.			Ref.			Ref.	
GG 2	1.8	1.11–3.08	0.018	3	0.82–11.09	0.09	1.1	0.58–1.75	0.9	1.2	0.29–5.4	0.8
GG 3	3.6	1.99–6.43	<0.001	10.6	2.71–41.42	0.0007	1.1	0.43–2.59	0.9	0.9	0.09–9.98	0.9
GG 4	2.7	1.45–5.03	0.002	8.1	1.93–33.81	0.004	0.9	0.39–2.18	0.9	3.2	0.69–14.94	0.1
GG 5	3.6	1.99–6.44	<0.001	31.9	8.08–126.2	<0.0001	2.9	1.39–6.03	0.005	5.8	1.29–25.7	0.02
LNI	2.1	1.37–3.22	0.0006	1.4	0.75–2.78	0.3	1.3	0.67–2.57	0.4	2	0.83–5.01	0.1
SM	1.7	1.22–2.25	0.0013	1.8	0.96–3.2	0.07	1.3	0.84–2.04	0.2	2.2	0.89–5.45	0.09

CI—confidence interval, PSA—prostate-specific antigen, LNI—lymph node invasion and SM—surgical margins.

**Table 3 cancers-13-02453-t003:** Multivariate competing risk analysis of the biochemical recurrence (BCR), developing metastases (MTS), overall mortality (OM) and cancer-specific mortality (CSM) in the low-, intermediate- and high-risk groups.

BCR				MTS			OM			CSM		
Parameter	HR	95% CI	*p*	HR	95% CI	*p*	HR	95% CI	*p*	HR	95% CI	*p*
Low Risk Group *												
PSA (ng/mL)	0.94	0.65–1.37	0.7	14.6	0.11−	0.2	1.3	0.88–1.9	0.1	0.9	0.33–2.23	0.7
Age (years)	1.1	0.97–1.26	0.1	1.1	0.62–1.65	0.9	1.1	0.95–1.18	0.2	0.8	0.6–1.18	0.3
Persistent PSA	3.9	0.79–19.84	0.09	21.4	0.1−	0.5	2.3	0.6–8.84	0.2	3.9	<0.0001−	0.9
SM	10.1	2.05–49.68	0.004	<0.1	<0.0001−	0.9	2.1	0.48–9.19	0.3	2.3	<0.0001−	0.9
Intermediate Risk Group **												
PSA (ng/mL)	1.1	0.99–1.11	0.1	0.96	0.82–1.1	0.6	0.9	0.91–1.07	0.8	1.2	0.95–1.51	0.1
Age (years)	1.1	0.99–1.06	0.2	1.1	0.94–1.14	0.5	1.1	1.05–1.19	0.002	1.2	0.93–1.61	0.2
Persistent PSA	3.8	2.16–6.77	<0.0001	2.6	0.74–9.4	0.1	1.1	0.42–2.5	0.9	4.2	0.48–36.11	0.2
Stage												
pT2		Ref.			Ref.			Ref.			Ref.	
pT3a	1.6	0.98–2.64	0.06	1.7	0.49–5.8	0.4	0.8	0.38–1.47	0.4	2.8	0.26–31.19	0.4
Grade Group												
GG 1		Ref.			Ref.			Ref.			Ref.	
GG 2		0.73–3.07	0.3	1	<0.0001−	0.9	0.8	0.38–1.73	0.6	1	<0.0001−	0.9
GG 3	2.3	0.98–5.6	0.06	1	<0.0001−	0.9	0.9	0.26–3.79	0.9	2.1	<0.0001−	0.9
SM	2.9	1.75–4.88	<0.0001	3.9	1.04–14.5	0.04	1.8	0.93–3.39	0.08	1.3	0.15–11.87	0.8
High Risk Group ***												
PSA (ng/mL)	1.1	0.99–1.02	0.5	0.9	0.97–1.01	0.5	0.9	0.96–1.02	0.4	0.9	0.91–1.01	0.08
Age (years)	1.1	0.98–1.03	0.8	1.1	0.96–1.06	0.7	1.1	1.01–1.12	0.015	1.1	0.99–1.15	0.09
Persistent PSA	5.1	3.31–7.99	<0.0001	2.6	1.2–5.62	0.015	2.7	1.36–5.45	0.005	6.2	1.66–23.06	0.007
Stage												
pT2		Ref.			Ref.			Ref.			Ref.	
pT3a	1.8	0.95–3.39	0.07	0.5	0.17–1.32	0.2	0.9	0.33–2.41	0.8	0.3	0.05–1.75	0.2
pT3b–pT4	2.3	1.2–4.24	0.01	1.1	0.42–2.55	0.9	2	0.83–4.83	0.1	1.9	0.56–6.6	0.3
Grade Group												
GG 1		Ref.			Ref.			Ref.			Ref.	
GG 2	4.1	0.92–18.23	0.06	1.6	0.18–15.09	0.7	0.8	0.19–2.99	0.7	0.7	0.06–9.04	0.8
GG 3	8.7	1.87–40.14	0.006	4.8	0.53–44.01	0.2	0.9	0.18–4.37	0.9	1.6	0.08–33.07	0.8
GG 4	5	1.12–22.02	0.035	3.9	0.43–35.07	0.2	0.8	0.2–3.21	0.8	2.7	0.25–30.59	0.4
GG 5	6.7	1.53–29.19	0.01	14.3	1.65–123.4	0.016	2.7	0.6–8.6	0.2	5.7	0.56–58.19	0.1
LNI	1.8	1.2–2.78	0.005	1.4	0.74–2.8	0.3	1.1	0.56–2.29	0.7	2.1	0.81–5.39	0.1
SM	1.1	0.77–1.65	0.5	1.6	0.8–3.35	0.2	0.8	0.4–1.48	0.4	1.9	0.66–5.77	0.2

CI—confidence interval, PSA—prostate-specific antigen, LNI—lymph node invasion and SM—surgical margins. * Multivariate analysis adjusted for preoperative PSA, age, surgical margin status and PSA persistence. ** Multivariate analysis adjusted for preoperative PSA, age, surgical margin status, stage pT2 vs. pT3a, grade group 1 vs. 2 vs.3 and PSA persistence. *** Multivariate analysis adjusted for preoperative PSA, age, surgical margin status, stage pT2 vs. pT3a, grade groups, LNI and PSA persistence.

## Data Availability

The data presented in this study are available within the article.

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
