# Peer review of "The Significance of Prostate Specific Antigen Persistence in Prostate Cancer Risk Groups on Long-Term Oncological Outcomes"

_cancers, 2021, doi:10.3390/cancers13102453_

Round 1

Reviewer 1 Report

I appreciate the efforts of the authors on this manuscript. It is very well written and organized.

Persistent PSA after radical prostatectomy for prostate cancer is a well-known parameter to assess the recurrence risk for local and distant metastatic disease. Multiple studies have looked into this and have come up with nomograms utilizing pre-op PSA, pathologic characteristics such as Gleason score, surgical margins, LN involvement at the time of surgery. EAU guidelines recommend utilizing PSA doubling time and Gleason score to risk-stratify patients to low risk and high risk.

The authors have focused on immediate post-surgery PSA persistence between 4-8 weeks and demonstrated that it predicted recurrence/relapse in low, intermediate and poor risk.

In my opinion, the serial PSA values with doubling time would be more accurate to utilize rather an a one point value from 4-8 weeks. Did the authors look at the doubling time as well to see if it correlated with outcomes.

Otherwise no significant questions about the study.

I appreciate the efforts of the authors on this manuscript. It is very well written and organized.

Persistent PSA after radical prostatectomy for prostate cancer is a well-known parameter to assess the recurrence risk for local and distant metastatic disease. Multiple studies have looked into this and have come up with nomograms utilizing pre-op PSA, pathologic characteristics such as Gleason score, surgical margins, LN involvement at the time of surgery. EAU guidelines recommend utilizing PSA doubling time and Gleason score to risk-stratify patients to low risk and high risk.

The authors have focused on immediate post-surgery PSA persistence between 4-8 weeks and demonstrated that it predicted recurrence/relapse in low, intermediate and poor risk.

In my opinion, the serial PSA values with doubling time would be more important to utilize rather an a one point value from 4-8 weeks. Did the authors look at the doubling time as well to see if it correlated with outcomes.

Otherwise no significant suggestions about the study.

Author Response

Dear Reviewer,

Thank you for positive rating of our manuscript. We agree that PSA doubling time could be very informative and possible significant factor. However, at this time, we have not such data and can answer to your raised question. That could be title for further our analyses. Thank you.

Sincerely,

Daimantas Milonas

Reviewer 2 Report

The authors wanted to test the impact of PSA persistence after prostatectomy on some oncologic outcomes (BCR, CSM, OM). More over they focused their efforts of dividing their population into three risk soubgroups (low, intermediate and high) according to the PCa pathological characteristics.

I fonud the paper well written, with a correct english form.

My concerns as follows:

- The authors included a very specific population of patients reporting a relatively high rate of positive surgical margins (PSM) per risk group - slightly higher compared o the available literature.

  • Indeed, It is a single center study with a consistent number of patients, for this reason i suppose that all the surgeons performing prostatectomies had passed the learning curve. For this reason, i would suggest to the authors to add to the manuscript how many surgeons took part to the 1200 surgeries and if they completly handled the procedure.
  • Due to the previous point, there is any correlation between the PSA persistence and the PSM compared to the R0 population?

Author Response

Dear Reviewer,

Thank you for positive assessment of our analyses. Indeed, in University hospital teaching process could influence the rate of PSM that is slightly higher in our analyses comparing to the literature. During study period, 9 senior urologists performed RP and we included this number in the manuscript.

Regarding influence of PSM on outcomes:

PSM was significant predictor of BCR in low and intermediate risk groups, while in high risk group other predictors were more important in multivariate analyses. In univariate analysis PSM was significant predictor. PSM was not significant predictor in other survival analyses except in intermediate risk group (significant predictor of metastases). In low risk group number of events (metastases, cancer related death) was low and analyses were inconclusive for this reason. In high risk group – other predictors were more important and significance of PSM reduced. However, in univariate analysis PSM is significant predictor fof metastases and cancer related death.

All this information is presented in tables 2 and 3.

Sincerely,

Daimantas Milonas